# Teacher Harassment Victimization in Adolescents with High-Functioning Autism Spectrum Disorder: Related Factors and Its Relationships with Emotional Problems

**DOI:** 10.3390/ijerph17114057

**Published:** 2020-06-06

**Authors:** Po-Chun Lin, Li-Yun Peng, Ray C. Hsiao, Wen-Jiun Chou, Cheng-Fang Yen

**Affiliations:** 1Department of Psychiatry, E-Da Hospital, Kaohsiung 82445, Taiwan; bernie0421@gmail.com; 2School of Medicine and Graduate Institute of Medicine, College of Medicine, Kaohsiung Medical University, Kaohsiung 80708, Taiwan; 3Department of Child and Adolescent Psychiatry, Jianan Psychiatric Center, Ministry of Health and Welfare, Tainan 71742, Taiwan; liyun619@msn.com; 4Department of Psychiatry and Behavioral Sciences, University of Washington School of Medicine, Seattle, WA 98195-6560, USA; rhsiao@u.washington.edu; 5Department of Psychiatry, Children’s Hospital and Regional Medical Center, Seattle, WA 98105, USA; 6College of Medicine, Chang Gung University, Taoyuan 33302, Taiwan; 7Department of Child and Adolescent Psychiatry, Chang Gung Memorial Hospital, Kaohsiung Medical Center, Kaohsiung 83301, Taiwan; 8Department of Psychiatry, Kaohsiung Medical University Hospital, Kaohsiung 80708, Taiwan

**Keywords:** teacher harassment, autism spectrum disorder, oppositional defiant disorder, social impairment, depression, anxiety, suicide

## Abstract

This study aimed to examine the prevalence, related factors, and emotional problems associated with teacher harassment victimization in adolescents with autism spectrum disorder (ASD) assessed by self-reports and parent reports. A total of 219 adolescents with ASD participated in this study. The self-reported and parent-reported rates of teacher harassment victimization were calculated. Sociodemographic characteristics, parent-reported social communication deficits, attention-deficit and hyperactivity disorder (ADHD) symptoms, oppositional defiant disorder (ODD) symptoms, self-reported depression and anxiety symptoms, and suicidality were surveyed. In total, 26 (11.9%) adolescents with ASD experienced teacher harassment based on self-reports or parent reports; the convergence between adolescent and parent reports on adolescent experiences of teacher harassment was low. Victims of teacher harassment exhibited more severe social communication deficits and ODD symptoms than nonvictims of teacher harassment. Victims of teacher harassment displayed more severe depression and anxiety and were more likely to have suicidality. Socio-communication deficits and ODD symptoms were related to teacher harassment victimization, which in turn was significantly associated with emotional problems among adolescents with ASD.

## 1. Introduction

### 1.1. Teacher Harassment and Its Negative Influences on Victimized Students

School violence is a serious social problem affecting the health and well-being of students. School violence is defined as any behavior intended to harm students or their property in a school [1]. Peer bullying and teacher maltreatment are critical types of school violence. Research has revealed that students who experience teacher maltreatment are more likely to develop severe mental and physical health problems, such as depression, anxiety, feelings of frustration or anger, desire for revenge [2,3], aggressive behaviors, dependency and regression, reexperiencing the trauma inflicted by the educator [2,4], and poor quality of life [5]. Teacher maltreatment also affects the student–teacher relationship. Victimized students may fear teachers rather than respect them; the teachers become aversive individuals in the students’ life [6].

Several types of teacher maltreatment have been examined, including physical, emotional, and sexual [6,7,8]. Teacher harassment is a type of emotional maltreatment. Numerous countries have implemented restrictions on corporal punishment in schools; however, harsh verbal discipline may turn into harassment. A study on 1376 adolescent students in Taiwan determined that 11.6% of students experienced emotional victimization, including mockery, insults, or humiliation or cursing [9]. Another study on 6160 adolescent students in Taiwan reported that 13.5% of students self-reported teacher harassment victimization [10]. Students who experience teacher harassment are at increased risk of developing poor self-esteem, somatic complaints [6], suicidal tendencies [11], and drug or alcohol abuse [12]. Mental health and educational professionals should develop prevention and intervention strategies for teacher harassment, such as providing preservice training about the management skills in the situations that can provoke teacher harassment, and information on the evidence showing the negative impacts of teacher harassment on students and its ineffectiveness in encouraging positive changes in students’ behaviors [6].

### 1.2. Teacher Harassment Victimization in Students with Autism Spectrum Disorder (ASD)

Autism spectrum disorder (ASD) is a neurodevelopmental disorder characterized by social communication impairments and restricted repetitive patterns of behaviors [13]. Children and adolescents with ASD are at a higher risk of peer bullying at school, with 19.2% of children with ASD often being victims of peer bullying compared with 4.6% for community children and 10.3% often involved in teasing others compared with 2.1% of community control children [14]. Several hypotheses have been made to explain the characteristics of children with ASD that heighten the risk of involvement in peer bullying at school [15,16,17]. For example, students with ASD often have problems with social cognition, which may result in a lack of appropriate skills for developing positive relationships [18,19]. Moreover, their peers may lack awareness and understanding, leading to reduced acceptance of differences [20], particularly concerning poor social and communicative skills [21] and the atypical behavioral traits [22] associated with ASD. However, to our knowledge, teacher harassment victimization in children and adolescents with ASD has not been examined. Poor interactions may also occur between students with ASD and their schoolteachers; thus, teacher harassment may be a challenge for students with ASD and require the attention of professionals in the fields of mental health and education.

### 1.3. Convergence of Parent-Student Reports on Teacher Harassment Victimization

The present study examined three problems related to teacher harassment victimization in adolescent students with ASD. The first problem is the convergence of parent–student reports on teacher harassment victimization. Research on the general adolescent population has revealed that adolescents may have different interpretations of interacting behaviors with peers compared to their teachers and parents [23]. The core symptoms of ASD mean that adolescents with ASD may have different awareness levels, definitions, and attributions of the experiences of bullying involvement compared with their parents [24]. They may also misinterpret bullying situations as nonbullying [25]. Whether the awareness of teacher harassment victimization was similar between self-reports by adolescent students with ASD and parents’ reports warrants further study.

### 1.4. Factors Related to Victimization Caused by Teacher Harassment

The second aim of the present study was to examine the factors related to victimization caused by teacher harassment in adolescent students with ASD. Determining the factors related to teacher harassment victimization is an essential step for developing programs for prevention, early detection, and effective intervention for students with ASD. For example, educators and school climates can detect the vulnerable students with ASD early and further provide them with more resources to improve their awareness and coping strategies to teacher harassment. Victimization caused by teacher harassment may be the result of interactions among multiple systems, including students with ASD, schoolteachers, students’ families, and school administrative units. The social-ecological theory provides an organizing framework for understanding vulnerability patterns within an individual’s environment and teacher harassment victimization among students with ASD [26]. Studies on typical development in children have determined that male students who had a low socioeconomic status [7,27,28] perceived poor student–teacher relationships and were involved with at-risk peers [5], and were at a significantly higher risk of teacher maltreatment. However, the factors related to teacher harassment victimization in adolescent students with ASD have not been examined.

### 1.5. Association between Teacher Harassment Victimization and Emotional Problems

As previously stated, students who have experienced teacher harassment have increased risks of poor self-esteem, somatic complaints, suicidal tendencies, and substance use [6,11,12]. The association between teacher harassment victimization and emotional problems in adolescent students with ASD warrants academic study.

### 1.6. Study Hypotheses

This aim of this study was to examine the convergence of self-reported and parent-reported teacher harassment victimization, the factors related to teacher harassment victimization and the relationships with emotional problems in adolescent students with ASD in Taiwan. Three hypotheses were constructed based on the study findings. First, we hypothesized that the convergence between adolescent-reported and parent-reported student harassment by teachers was low. Second, we hypothesized that certain individual factors (adolescents’ demographic characteristics, ASD-related symptoms, inattention, hyperactivity and impulsivity, and oppositional defiant symptoms) and parental factors (parental educational level and socioeconomic status) were related to teacher harassment victimization. Third, we hypothesized that teacher harassment victimization was associated with depression, anxiety, and suicidality in adolescent students with ASD.

## 2. Methods

### 2.1. Participants

Participants were recruited at five child psychiatry outpatient clinics in Taiwan, comprising one regional teaching hospital, one child psychiatry specified clinic, and three university-affiliated teaching hospitals. Taiwan’s National Health Insurance program allows patients to visit outpatient clinics of teaching hospitals without being transferred by general practitioners. Therefore, the adolescents enrolled from these five child psychiatry outpatient clinics were representative of the adolescent population of Taiwan. The inclusion criteria were as follows: (1) age of 11–18 years; (2) diagnosis of ASD according to the fifth edition of the Diagnostic and Statistical Manual of Mental Disorders (DSM-5; American Psychiatric Association, 2013); (3) full-scale intelligence quotient (FSIQ) ≥80, determined using the Chinese version of the Wechsler Intelligence Scale for Children, fourth edition [29]; (4) verbal communication ability; and (5) currently studying in an inclusive classroom, not a special education classroom. Participants who satisfied the criteria for inclusion were consecutively recruited between August 2013 and July 2016. Participants whose parents had an intellectual disability, bipolar disorder, schizophrenia, or any cognitive deficits that resulted in significant community difficulties were excluded. A total of 228 adolescents with high-functioning ASD were invited to participate, and 219 (96.1%) adolescents and their parents agreed to participate and were assessed by the research assistants based on the research questionnaire. All participants studied in standard classes without support. The Institutional Review Board of Kaohsiung Medical University (KMUHIRB-20120084) approved this study.

### 2.2. Measures

#### 2.2.1. Teacher Harassment Victimization

The experience of being harassed by teachers during the past year was assessed by the question: “Is there a teacher who picks on you on purpose?” Participants answered the question on a 4-point Likert scale from 0 (*never*) to 3 (*all the time*). Participants who answered *often* or *all the time* to this question were classified as having experienced teacher harassment [10].

#### 2.2.2. Chinese Social Responsiveness Scale

The parent-reported Chinese version of the Social Responsiveness Scale (SRS) contains 60 items evaluated on a 4-point Likert scale that assesses adolescents’ reciprocal social behaviors [30,31]. The Chinese version of the SRS is composed of four subscales: socio-communication, autism mannerisms, social awareness, and social emotion. A higher total score on the subscale indicates greater deficits in social responsiveness. Research has determined that the SRS effectively distinguishes between children and adolescents with and without ASD [30,31].

#### 2.2.3. Short Version of the Chinese Swanson, Nolan, and Pelham Version IV Scale

The short version of the Chinese Swanson, Nolan, and Pelham Version IV Scale (SNAP-IV) contains 26 items comprising the core DSM-IV-derived attention-deficit and hyperactivity disorder (ADHD) subscales of inattention, hyperactivity, and impulsivity, and oppositional symptoms [32,33]. Each item was evaluated on a 4-point Likert scale from 0 (*not at all*) to 3 (*very much*). A higher total score of the subscales indicates a greater level of ADHD and oppositional symptoms. In this study, the Cronbach’s α values for the inattention, hyperactivity and impulsivity, and oppositional subscales were 0.91, 0.91, and 0.92, respectively.

#### 2.2.4. Taiwanese Version of the Center for Epidemiological Studies Depression Scale

The adolescent-reported Taiwanese Version of the Center for Epidemiological Studies Depression (T-CES-D) scale comprises 20 items rated on a 4-point Likert scale from 1 (*rarely or none of the time*) to 4 (*most or all of the time*) that assess the frequency of depressive symptoms over the previous month [34,35]. A higher total score on the T-CES-D scale indicates more severe depression. In this study, the Cronbach’s α of the T-CES-D scale was 0.88.

#### 2.2.5. Taiwanese Version of the Multidimensional Anxiety Scale for Children

The adolescent-reported Taiwanese Version of the Multidimensional Anxiety Scale for Children (MASC-T) comprises 39 items that evaluate the level of anxiety symptoms over the previous month on a 4-point Likert scale [36,37]. A higher total score on the MASC-T indicates more severe anxiety symptoms. A study demonstrated that the MASC-T has acceptable reliability and validity [37]. In the present study, the Cronbach’s α for the MASC-T was 0.88.

#### 2.2.6. Suicidality

The adolescent-reported suicidality module of the epidemiological version of the Kiddie Schedule for Affective Disorders and Schizophrenia [38] contains five self-reported items that elicit a yes or no response to assess four forms of suicidal ideation and the occurrence of suicide attempts during the previous year [39]. A study reported that the Cohen’s kappa coefficient of agreement (κ) between adolescents’ self-reported suicidality and their parents’ reports was 0.541 (*p* < 0.001) [39]. The Cronbach’s α for the questionnaire on suicidality was 0.79. In the present study, participants who answered “yes” to any item were classified as having suicidality.

#### 2.2.7. Sociodemographic Characteristics

We examined the participants’ sex, age, parental education duration, and parental occupational socioeconomic status (SES). The levels of parental occupational SES were assessed using the Close-Ended Questionnaire of the Occupational Survey (CEQ-OS), which classifies paternal and maternal occupational SES into five levels [40]. A high level indicates a high occupational SES. The CEQ-OS has been demonstrated to have high reliability and validity and is frequently used in studies on children and adolescents in Taiwan [40]. In the present study, levels 1–3 and 4–5 were classified as low and high occupational SES, respectively.

### 2.3. Procedure

Adolescents with high-functioning ASD and their parents were invited to complete the research questionnaires. Adolescents’ self-reported experience of teacher harassment, depression, anxiety, and suicidality were collected by interviews conducted by research assistants. The parents completed questionnaires regarding adolescents’ experience of teacher harassment, adolescents’ deficits in social responsiveness, adolescents’ ADHD and ODD symptoms, and their demographic characteristics. Research assistants were available to assist in completing the questionnaires. Data analysis was performed using SPSS 20.0 statistical software.

### 2.4. Statistical Analysis

The prevalence of self-reported and parent-reported teacher harassment was calculated using descriptive statistics. Category variables (sex, parental occupational SES, suicidality) and continuous variables (age, parental education duration, deficits in social responsiveness, ADHD and ODD symptoms, depression and anxiety symptoms) were compared between the adolescents who did and did not experience teacher harassment using chi-square and t tests, respectively. Factors with a *p* value below 0.05 were included in a multivariable forward logistic regression analysis to examining their association with the experience of teacher harassment after adjusting for the effects of other variables. An odds ratio (OR) and its 95% confidence interval (CI) were used to represent the statistical significance.

The associations between teacher harassment victimization (independent variables) and depression, anxiety, and suicidality (dependent variables) were also examined, first by using t and chi-square tests, then by using multiple and logistic regression analysis models, in which the effects of sociodemographic characteristics were controlled for. A two-tailed *p* value of less than 0.05 was considered statistically significant.

## 3. Results

### 3.1. Prevalence of Teacher Harassment Victimization and Convergence of Adolescent–Parent Reports

Teacher harassment in the previous year was reported by 17 (7.8%) adolescents and 17 (7.8%) parents of adolescents. However, fewer than half of these cases (8 adolescents, 47.1%) were both self-reported and parent-reported cases of teacher harassment. In total, 26 (11.9%) adolescents experienced teacher harassment based on adolescent or parent reports.

### 3.2. Factors Related to Teacher Harassment Victimization

Table 1 summarizes the results of the comparisons between victims and nonvictims of teacher harassment regarding sociodemographic characteristics, deficits in social responsiveness, ADHD and ODD symptoms, depression, anxiety, and suicidality. Victims of teacher harassment had higher maternal occupational SES (χ^2^ = 5.598, *p* = 0.018), more severe deficits in the domains of social communication (*t* = −3.074, *p* = 0.002) and autism mannerism (*t* = −2.684, *p* = 0.008) on the SRS, more severe inattention (*t* = −2.817, *p* = 0.005), hyperactivity/impulsivity (*t* = −3.453, *p* = 0.001), oppositional defiant symptoms (*t* = −3.652, *p* < 0.001) compared with nonvictims of teacher harassment. These significant variables were then included in a multivariate forward logistic regression to further examine their association with being a victim of teacher harassment (Table 2). The results indicated that victims of teacher harassment had more severe deficits in social communication and ODD symptoms than nonvictims.

### 3.3. Teacher Harassment Victimization and Emotional Problems

Table 1 also reports differences in the levels of depression and anxiety and the proportion of adolescents with suicidality between victims and nonvictims of teacher harassment. The results indicated that victims of teacher harassment had more severe depression and anxiety than nonvictims of teacher harassment. Victims of teacher harassment were at higher risk of suicidality than nonvictims. The associations between being a victim of teacher harassment and depression, anxiety, and suicidality were further examined using multiple and logistic regression analysis models to control for sociodemographic characteristics. The victims of teacher harassment exhibited more severe depression (beta = 0.264, *t* = 4.152, *p* < 0.001) and anxiety (beta = 0.137, *t* = 2.036, *p* = 0.043) than the nonvictims. Victims of teacher harassment were also more likely to have suicidality (OR = 6.848, 95% CI of OR: 2.708–17.316) than nonvictims.

## 4. Discussion

The present study revealed that 11.9% of students with ASD experienced self-reported or parent-reported teacher harassment victimization during the previous year, whereas fewer than half of the cases (47.1%) were both self-reported and parent-reported. Adolescent victims of teacher harassment had more severe social communication deficits and ODD symptoms than the nonvictims of teacher harassment. Furthermore, victims of teacher harassment had more severe depression and anxiety and a higher risk of suicidality than nonvictims of teacher harassment.

### 4.1. Prevalence of Teacher Harassment Victimization

One previous study on 1376 adolescent students in Taiwan found that 11.6% of the sample experienced emotional victimization [9]. The rate of teacher harassment victimization among adolescent students with ASD observed in the present study (11.9%) was similar to those reported by studies on the general adolescent population [9,10]. The core symptoms of ASD, including social communication impairments and restricted repetitive patterns of behaviors, may increase the difficulty of student–teacher communication and teachers’ classroom management, which led us to develop the hypothesis that teacher harassment victimization may be higher among adolescent students with ASD compared with the general adolescent population. However, the result of the present study did not support this hypothesis. Further study is warranted to examine whether the prevalence of teacher harassment victimization in students with ASD varies depending on the severity of ASD, sources, informants, and survey instruments.

### 4.2. Hypotheses Vertification

#### 4.2.1. Low Convergence of Adolescent–Parent Reports

This study further revealed that adolescents with ASD and their parents had low convergence in reporting adolescents’ teacher harassment victimization. Several reasons may account for this result. First, whether adolescents with ASD could perceive teacher harassment correctly is questionable. Baron-Cohen et al. [41] have proposed that individuals with ASD have deficits in the theory of mind (ToM) task performance, which results in difficulty in understanding others’ viewpoints and leads to misinterpretation of bullying situations [42]. However, other studies have challenged the association between deficits in ToM task performance and the misidentification of bullying [43,44,45]. For instance, Bitsika and Sharpley [43] revealed that boys with high-functioning ASD aged 7–12 years could identify bullying behaviors. Second, parents might not be aware of their child’s experience of victimization [46]. Adolescents with ASD may not inform their parents of teacher harassment victimization because they felt ashamed, they thought that it was not serious enough, they were worried that informing their parents may worsen the problems, or they attempted to solve the problems themselves. Third, people in Taiwan are influenced by traditional Confucianism and thus usually respect teachers and consider that they have higher power than students in the hierarchical relationship [47]. ASD features may also complicate adolescents’ communication and interaction with their parents. Parents may interpret teacher harassment in a positive manner and neglect students’ feelings. Moreover, ASD has a considerable hereditary etiology [48], and parents of children with ASD may also exhibit autistic traits, further hindering communication between adolescents and parents.

Despite the low agreement regarding teacher harassment victimization between the adolescent reports and the parent reports, parents play a vital role in preventing and managing bullying behaviors [49], including teacher harassment. The results of this study serve as a reminder of the importance of cultivating high-quality parent–child relationships among adolescents with ASD to improve their communication and enable parents to understand their children’s experience of teaching harassment. Mental health and educational professionals should consider both adolescent and parent reports when gathering information regarding teacher harassment victimization in adolescents with ASD. Third party information from classmates or other teachers could aid in understanding the situation more comprehensively.

#### 4.2.2. Factors Related to Teacher Harassment Victimization

This study demonstrated that adolescent victims of teacher harassment had more severe social communication deficits and ODD symptoms than nonvictims of teacher harassment. Research has revealed that poor student–teacher relationships were significantly associated with teacher maltreatment in the general adolescent student population [50]. The deficit in social responsiveness among students with ASD may worsen student–teacher relationships and increase the risk of teacher harassment victimization.

Children and adolescents presenting ODD symptoms may have negativistic, disobedient, and hostile behaviors toward authority figures, as well as an inability to assume responsibility for their mistakes, leading them to blame others. These characteristics may increase students’ social difficulties with their peers and teachers [48]. The uncooperative behavior of adolescent students with ASD and comorbid ODD may be difficult for teachers to manage. Consequently, teachers may feel stressed and overwhelmed by students’ behavioral problems in the classroom [50]. Agnew’s general strain theory [51] indicates that numerous aggressive acts are explained by strain that causes the use of aggressive behavior to reduce or escape from the strain. Therefore, teachers that lack resources and support to manage strain may be at higher risk of maltreating their students when managing students’ behavioral problems [52]. Therefore, training and supporting teachers to cope with their strain and handle disobedient behaviors in adolescent students with ASD and comorbid ODD symptoms are vital preventive strategies of teacher harassment.

The results of the present study indicate that the victims of teacher harassment had a higher maternal occupational SES than nonvictims in the chi-square test, although the difference was nonsignificant in the multivariate logistic regression analysis. The finding did not accord with studies on the general student population in the West and Middle East that identified low family SES as a risk factor of teacher maltreatment victimization [1,7,27,28]. Theoretically, students from poor families are more likely to experience victimization because their relatively limited family resources and power may designate them as targets of violence [27,28]. However, a study in Taiwan reported no significant association between family SES and school violence among students [53]. Some researchers argued that compared with Western countries, family income distribution was relatively equal in Taiwan, and thus the association between family SES and school violence may be less significant [53,54]. We hypothesized that mothers with a high occupational SES may be more demanding of teachers, which may increase teachers’ stress and the subsequent risk of verbal violence and harassment directed at students with ASD. Furthermore, mothers with high occupational SES may be more prone to detect and report teachers’ harassment. However, these hypotheses require further examination.

The chi-square test demonstrated that victims of teacher harassment had more severe ADHD symptoms than nonvictims, although the difference was nonsignificant in the multivariate logistic regression analysis. ADHD symptoms, including inattention, hyperactivity, and impulsivity, may increase teachers’ burden in classrooms; therefore, teachers may have negative reactions toward students with ASD who have severe ADHD symptoms. Furthermore, research on children and adolescents with ASD revealed that comorbid ADHD was significantly associated with peer bullying victimization [55]. Teacher harassment victimization should be routinely surveyed in adolescents with ASD and comorbid ADHD.

#### 4.2.3. Teacher Harassment Victimization and Emotional Problems

The present study determined that teacher harassment victims with ASD had more severe depression and anxiety and were more likely to have suicidality than ASD nonvictims. The results of the present study accorded with studies on the general student population [1,6,11,56]. The cross-sectional study design limited the possibility to determine the causal relationship between teacher harassment victimization and emotional problems, but the results supported the contention that both teacher harassment victimization and emotional problems should be routinely surveyed in adolescent students with ASD.

### 4.3. Limitations

This study had several limitations. First, because of the cross-sectional study design, we could not reach conclusions on the temporal relationships between teacher harassment victimization and emotional problems of adolescents with ASD. Second, we used a single question to evaluate teacher harassment victimization. We did not survey the types and frequency of teacher harassment or the situation in which the teacher harassment occurred. Third, we did not obtain reports from a third party to determine the accuracy of self-reports and parent-reports on teacher harassment. Fourth, we did not examine certain potential confounders, such as parental mental health status, that may influence parental reports on their children’s experiences of teacher harassment. Fifth, adolescents with high-functioning ASD had FSIQs of 80 or higher, had verbal communication ability, and studied in inclusive classrooms. The results of the present study may not be generalizable to adolescents with ASD who have a low FSIQ, no verbal communication ability, or study in a specialized education room.

## 5. Conclusions

The present study identified a low agreement between self-reported and parent-reported teacher harassment victimization in adolescents with ASD. Mental health and educational professionals should collect both self-reported and parent-reported information to comprehensively detect teacher harassment victimization in adolescents with ASD and enhance their awareness of teacher harassment, and they should report such situations to parents or others. Deficits in socio-communication and ODD symptoms were significantly associated with teacher harassment victimization. Programs for enhancing socio-communicative ability are required to reduce the risk of adolescents with ASD being victimized by teacher harassment. Tailored training programs for teachers and non-teaching staff in schools to introduce them to the specifics of students with ASD and ODD and educate the appropriate strategies to promote their cooperation are also important. The risk of experiencing teacher harassment in adolescents with ASD and comorbid high-level ODD symptoms should be lowered through early intervention. Teacher harassment victimization was significantly associated with emotional problems in adolescent students with ASD. Mental health and educational professionals should be sensitive to the victimization and emotional problems related to teacher harassment in adolescents with ASD, cooperate to detect and monitor the situation of teacher harassment, and provide necessary assistance.

## Figures and Tables

**Table 1 ijerph-17-04057-t001:** Comparisons of sociodemographic characteristics, deficits in social responsiveness, ADHD and ODD symptoms, depression, anxiety, and suicidality between victims and non-victims of teacher harassment (N = 219).

Variable	Non-Victims (*n* = 193) *n* (%)	Victims (*n* = 26) *n* (%)	χ^2^	*t*	*p*
Sex, *n* (%)					
Girls	24 (12.4)	3 (11.5)	0.017		0.896
Boys	169 (87.6)	23 (88.5)			
Age (years), mean (SD)	13.7 (2.1)	14.1 (2.2)		−1.015	0.311
Paternal education duration (years), mean (SD)	14.6 (2.9)	15.3 (2.8)		−1.159	0.248
Maternal education duration (years), mean (SD)	14.2 (2.5)	14.3 (2.5)		−0.250	0.803
Paternal occupational SES, *n* (%)					
High	111 (57.5)	18 (69.2)	1.300		0.254
Low	82 (42.5)	8 (30.8)			
Maternal occupational SES, *n* (%)					
High	72 (37.3)	16 (61.5)	5.598		0.018
Low	121 (62.7)	10 (38.5)			
Deficits in social responsiveness on the SRS, mean (SD)					
Socio-communication	67.4 (13.6)	76.3 (15.1)		−3.074	0.002
Autism mannerism	33.3 (7.2)	37.4 (8.7)		−2.684	0.008
Social awareness	31.3 (4.6)	33.1 (4.9)		−1.937	0.054
Social emotion	20.5 (3.7)	23.0 (4.9)		−1.819	0.079
ADHD and ODD symptoms on the SNAP-IV, mean (SD)					
Inattention	14.2 (6.3)	18.0 (7.0)		−2.817	0.005
Hyperactivity/impulsivity	9.4 (6.3)	14.1 (8.5)		−3.453	0.001
Oppositional defiant	10.2 (5.9)	14.8 (7.3)		−3.652	<0.001
Depression on the CESD, mean (SD)	13.6 (9.3)	21.9 (12.8)		−4.053	<0.001
Anxiety symptoms on the MASC-T, mean (SD)	34.4 (16.3)	41.5 (19.0)		−2.062	0.040
Suicidality, *n* (%)					
No	162 (83.9)	12 (46.2)	20.037		<0.001
Yes	31 (16.1)	14 (53.8)			

ADHD: attention-deficit hyperactivity disorder; MASC-T: Taiwanese version of the Multidimensional Anxiety Scale for Children; CESD: the Center for Epidemiological Studies-Depression Scale; ODD: oppositional defiant disorder; SES: socioeconomic status; SNAP-IV: Swanson, Nolan, and Pelham, Version IV Scale; SRS: Social Responsiveness Scale.

**Table 2 ijerph-17-04057-t002:** Factors related to the victims of teacher harassment: forward logistic regression analysis.

Variable	Victims of Teacher Harassment	
	OR	95% CI of OR
Socio-communication	1.144	1.010–1.297
Oppositional defiant symptoms	1.101	1.025–1.182

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
