# Peer review of "Teacher Harassment Victimization in Adolescents with High-Functioning Autism Spectrum Disorder: Related Factors and Its Relationships with Emotional Problems"

_ijerph, 2020, doi:10.3390/ijerph17114057_

Round 1

Reviewer 1 Report

Dear Authors!

Thank you for taking up a very important topic of Teacher Harassment Victimization in Adolescents With High-Functioning Autism Spectrum Disorder.

In my opinion, the paper is valuable and presents high quality both concerning research design, data analysis, presentation of results, as well as conclusions.

The list of shortcomings to be removed is very short and presented below:

1) Part of the text in the „Discussion” section referring to the authors' previous studies (line 260-263) is a repetition of the text from part 1.1 (lines 66-69). There is no need to repeat it.

2) Paragraph 4.1 „Prevalence of teacher harassment victimization” (line 259) in my opinion is redundant as a paragraph and should be just a part of the discussion section. As three hypotheses were set, three paragraphs (each for single hypothesis should appear in the discussion section, plus the „Limitations” section), so the structure would be as follows:

4. Discussion
4.1. Low convergence of adolescent–parent reports
4.2. Factors related to teacher harassment victimization
4.3. Teacher harassment victimization and emotional problems
4.4. Limitations

OR

If paragraph 4.1 stays, I would consider adding another subtitle 4.2. (about verification of hypotheses) which would have three subparagraphs included, so the structure would be as follows:

4. Discussion
4.1. Prevalence of teacher harassment victimization
4.2. Hypotheses verification
4.2.1. Low convergence of adolescent–parent reports
4.2.2. Factors related to teacher harassment victimization
4.2.3. Teacher harassment victimization and emotional problems
4.3. Limitations

These are options to choose from. Please decide which way is closer to you. As a result, this very interesting paper will be easier to read.

3) The authors emphasize the need for prevention and early intervention to avoid the phenomenon of teacher harassment victimization (lines 71-72, 101-102, 372) but they do not propose any solutions under these interventions.
Maybe it is worth to mention what types of interventions would be the most appropriate in their opinion (e.g. tailored training programs for teaching and non-teaching staff in schools to introduce the specifics of working with children with ASD; some monitoring solutions for schools to detect the situation of teacher harassment victimization; making children with ASD aware of the signs of harassment and informing them about the need to report any such cases e.t.c.).

After removing a few indicated defects, I am happy to recommend the text for publication, as a valuable contribution to an important topic of support for the therapy of individuals with ASD.

Best regards,
The reviewer.

Author Response

Comment

  • Part of the text in the „Discussion” section referring to the authors' previous studies (line 260-263) is a repetition of the text from part 1.1 (lines 66-69). There is no need to repeat it.

Response

Thank you for your reminding. As your opinion, we deleted the repetition (line 260-263), and next we changed to “one previous study…”. In line 276, we also added reference 10 “[9.10]”.

Comment

2) Paragraph 4.1 „Prevalence of teacher harassment victimization” (line 259) in my opinion is redundant as a paragraph and should be just a part of the discussion section. As three hypotheses were set, three paragraphs (each for single hypothesis should appear in the discussion section, plus the „Limitations” section), so the structure would be as follows:

  1. Discussion
    4.1. Low convergence of adolescent–parent reports
    4.2. Factors related to teacher harassment victimization
    4.3. Teacher harassment victimization and emotional problems
    4.4. Limitations

OR

If paragraph 4.1 stays, I would consider adding another subtitle 4.2. (about verification of hypotheses) which would have three subparagraphs included, so the structure would be as follows:

  1. Discussion
    4.1. Prevalence of teacher harassment victimization
    4.2. Hypotheses verification
    4.2.1. Low convergence of adolescent–parent reports
    4.2.2. Factors related to teacher harassment victimization
    4.2.3. Teacher harassment victimization and emotional problems
    4.3. Limitations

These are options to choose from. Please decide which way is closer to you. As a result, this very interesting paper will be easier to read.

Response

Thank you for your reminding. We chose your second suggestion that adding another subtitle 4.2. (about verification of hypotheses). We changed the structure as the follows. Please refer to line 284, 285, 314, 356, 364.

“4. Discussion
4.1. Prevalence of teacher harassment victimization
4.2. Hypotheses verification
4.2.1. Low convergence of adolescent–parent reports
4.2.2. Factors related to teacher harassment victimization
4.2.3. Teacher harassment victimization and emotional problems
4.3. Limitations”

Comment

3) The authors emphasize the need for prevention and early intervention to avoid the phenomenon of teacher harassment victimization (lines 71-72, 101-102, 372) but they do not propose any solutions under these interventions.
Maybe it is worth to mention what types of interventions would be the most appropriate in their opinion (e.g. tailored training programs for teaching and non-teaching staff in schools to introduce the specifics of working with children with ASD; some monitoring solutions for schools to detect the situation of teacher harassment victimization; making children with ASD aware of the signs of harassment and informing them about the need to report any such cases e.t.c.).

Response

Thank you for your reminding. In the revised manuscript, we added actual prevention and intervention to avoid the phenomenon of teacher harassment in Introduction and Conclusion section.

“…, such as providing the preservice training about the management skills to the situations that can provoke teacher harassment, and information on the evidence showing the negative impacts of teacher harassment to students and its ineffectiveness in bringing positive changes in students’ behaviors [6].” Please refer to line 72-75.

“For example, educators and school climates can early detect the vulnerable students with ASD, and further provide them more resources to improve their awareness and coping strategies to teacher harassment.” Please refer to line 106-108.

“and enhance their awareness of teacher harassment and they will report such situations to parents or others.” Please refer to line 381-382.

“Tailored training programs for teacher and non-teaching staff in school to introduce the specifics of students with ASD and ODD and educate the appropriate strategies to promote their cooperation are also important.” Please refer to line 385-387.

“cooperate to detect and monitor the situation of teacher harassment.” Please refer to line 392-393.

Reviewer 2 Report

Hello and thank you for your study.

I add file.

Kind regards.

Author Response

Comment

I agree with the limitations of the study because more details would have to be entered. People with high-level autism may have special high sensitivity in the face of a reprimand or correction and interpret it as teacher’s aggression. We know that they have little ability to manage frustration. On the other hand, there may also be overprotection on the part of the parents, which would justify the difference between the report of the teachers and that of the parents. In any case, the results are not very clear. What parameters have been compared with the chi square? What are the results of the t test? Excuse me if I haven't seen you.

Response

Thank you for your reminding. In the revised manuscript we added explanation for the use of chi square and t tests in Methods section, as well as we added the results of chi square and t tests in Results section as below. We also separated the column 4 “χ2 or t” into column 4 “χ2” and column 5 “t”.

“Category variables (sex, parental occupational SES, suicidality) and continuous variables (age, parental education duration, deficits in social responsiveness, ADHD and ODD symptoms, depression and anxiety symptoms) were compared between the adolescents who did and did not experience teacher harassment using chi-square and t tests, respectively.” Please refer to line 215-218.

“Victims of teacher harassment had higher maternal occupational SES (χ2 =5.598, p =0.018), more severe deficits in the domains of social communication (t =-3.074, p =0.002) and autism mannerism (t =-2.684, p =0.008) on the SRS, and more severe inattention (t =-2.817, p =0.005), hyperactivity/impulsivity (t =-3.453, p =0.001), oppositional defiant symptoms (t =-3.652, p <0.001) compared with nonvictims of teacher harassment.” Please refer to line 237-241.